# Undrained Behavior of Microbially Induced Calcite Precipitated Sand with Polyvinyl Alcohol Fiber

**Sun-Gyu Choi [1], Tung Hoang [2,3] and Sung-Sik Park [4],***

[1] Postdoctoral Researcher, Civil, Construction and Environmental Engineering, Korea Advanced Institute of Science and Technology (KAIST), 291 Daehak-ro, Yuseong-gu, Daejeon 34141, Korea; choisg@kaist.ac.kr

[2] Graduate Student, Department of Civil, Construction, and Environmental Engineering, Iowa State University, Ames, IA 50011, USA; tphoang@iastate.edu

[3] Faculty of Bridge and Road Construction Engineering, The University of Danang—University of Science and Technology, Danang 550000, Vietnam

[4] Department of Civil Engineering, Kyungpook National University, 80 Daehakro, Bukgu, Daegu 41566, Korea

* Correspondence: sungpark@knu.ac.kr; Tel.: +82-53-950-7544

**Abstract:** Microbially induced calcite precipitation can cement sand and is an environment-friendly alternative to ordinary Portland cement. In this study, clean Ottawa sand was microbially treated to induce calcite contents (CCs) of 0%, 2%, and 4%. Polyvinyl alcohol fiber was also mixed with the sand at four different contents (0%, 0.2%, 0.4%, and 0.6%) with a constant CC of 4%. A series of undrained triaxial tests was conducted on the treated sands to evaluate the effects of the calcite treatment and fiber inclusion. Their hydraulic conductivity was also determined using a constant head test. As the CC increased from 0% to 4%, the friction angle and cohesion increased from 35.3° to 39.6° and from 0 to 93 kPa, respectively. For specimens with a CC of 4%, as the fiber content increased from 0% to 0.6%, the friction angle and cohesion increased from 39.6° to 42.8° and from 93 to 139 kPa, respectively. The hydraulic conductivity of clean Ottawa sand decreased by a factor of more than 100 as the CC increased from 0% to 4%. The fiber inclusion had less effect on the hydraulic conductivity of the specimen with 4% CC.

**Keywords:** microbially induced calcite precipitation; polyvinyl alcohol fiber; fiction angle; cohesion; hydraulic conductivity

## 1. Introduction

Microbially induced calcite precipitation (MICP) has received considerable attention in recent years because it has the potential to replace ordinary Portland cement in geotechnical stabilization practices. In addition, significant environmental benefits might be achieved by reducing the carbon dioxide ($CO_2$) otherwise required for manufacturing cement. MICP processing with ureolytic bacteria is a process based on the use of enzyme ureases and complementary chemicals (typically urea and calcium chloride). This is illustrated by the following reactions [1]:

Reaction 1: $CO(NH_2)_2 + 2H_2O \rightarrow 2NH_4^+ + CO_3^{2-}$
Reaction 2: $CO_2 + H_2O \rightarrow H_2CO_3$
Reaction 3: $Ca^{2+} + CO_3^{2-} \rightarrow CaCO_3$

Intent on promoting pragmatic geotechnical applications of these biochemical conversions, a number of researchers [2–5] have published studies on biocementation using MICP methods over the past two decades. At the same time, the engineering properties of MICP biocementation have also been investigated on multiple levels. For example, several investigations [6–8] have confirmed that

the unconfined compressive strength (UCS) increases with the MICP-derived calcite content (CC). However, it has also been found that the hydraulic conductivity decreases as the CC increases [5,9,10]. Other researchers have evaluated biocementation using shear wave velocity [2,11,12], while yet another group of researchers have confirmed the levels of biocementation using triaxial [12,13] and dynamic testing [1,14]. In addition to MICP treatment, artificial fiber can be combined to improve the physical properties beyond those achieved with either cement-based processing (which typically shows brittleness and low strength [15]) or standard MICP treatment (which similarly tends to show brittleness [16]).

Li et al. [17] recently published an article that addresses a similar addition of co-blended fibers to improve the MICP performance of sand biocementation, and reported on the use of homopolymer polypropylene multifilament fiber. This addition showed that suitable engineering properties were obtained with fiber additions of up to 0.3% fiber content by weight of sand. Choi et al. [18] also reported similar MICP enhancements using polyvinyl alcohol (PVA) fiber. Their work revealed that the addition of PVA fiber increased the UCS and tensile strength of the MICP-treated sand by 30% and 160%, respectively. However, despite the various analyses of the engineering properties of biocementations, biocementations with fiber are not clearly understood.

In this study, two different effects, calcite precipitation and fiber inclusion, were investigated in order to understand the engineering properties of biocementation by the MICP method using triaxial and permeability tests. The effect of calcite precipitation was evaluated for different CCs (0%, 2%, and 4%). The effect of fiber inclusion was compared for specimens with four different fiber contents (0%, 0.2%, 0.4%, and 0.6%) and the same CC (4.0%).

## 2. Sample Preparation

### 2.1. Sand and Fiber

Ottawa sand, as described in ASTM C778-17 [19], was used as the standard treatment material for this study. This sand has grain sizes ranging from 0.6 to 0.85 mm with a mean grain size of 0.73 mm, as shown in Figure 1a. Ottawa sand has a specific gravity of 2.65 and it is classified as SP according to the Unified Soil Classification System. The maximum and minimum void ratios are 1.1 and 0.6, respectively.

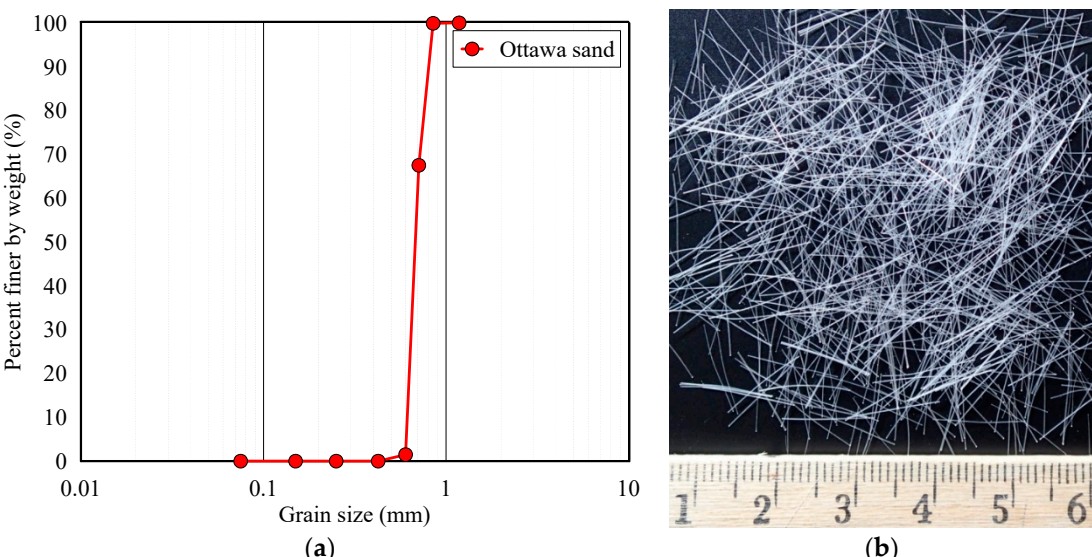

**Figure 1.** (**a**) Grain size distribution curve of Ottawa sand and (**b**) the polyvinyl alcohol (PVA) fibers used in the tests.

PVA fibers shown in Figure 1b were used. They have a pre-cut form with a typical diameter and length of 0.1 and 12 mm, respectively. The same type of PVA fibers are known to provide similar engineering benefits when applied in cement and in cemented sand processing [20] as well as with MICP-biocemented sands [18]. Table 1 shows the physical properties for this form of pre-cut PVA fiber.

**Table 1.** Properties of the polyvinyl alcohol (PVA) fiber.

| Type of PVA Fiber (Kuraray, Japan) | Specific Gravity | Cut Length (mm) | Diameter (mm) | Tensile Strength (MPa) | Young's Modulus (MPa) |
|---|---|---|---|---|---|
| RECS 100L | 1.3 | 12 | 0.1 | 1078 | 25,000 |

### 2.2. Bacterial Cultivation

*Sporosarcina pasteurii* (American Type Culture Collection, ATCC 11859) was used as the urease-producing bacteria (UPB) [2,9]. The chemical applied during the amended MICP treatment procedure included calcium chloride and urea with a chemical concentration of 0.3 M. Yeast extract (20 g/L) and ammonium sulfate (($NH_4)_2SO_4$) (10 g) mixed into 0.13 M of tris buffer (pH = 9.0) solution was used as a medium [7]. The prepared medium was sterilized in an autoclave at 121 °C for 15 min and then cultured at 30 °C in a shake-table incubator for two days. The optical density (OD) of the culture was then measured to ensure that it was in the range of 1.2 to 1.6.

### 2.3. Sample Preparation and Treatment

Two types of sand columns were prepared for biocementation with and without fiber. In the case without fiber, the Ottawa sand was placed into a PVC mold to make a dry unit weight of 16.19 kN/m$^3$ (60% of relative density). In the case with fiber, the Ottawa sand was first placed in a Hobart mixer where it could be mixed homogenously with the PVA fiber. By weight, 0%, 0.2%, 0.4%, and 0.6% PVA fiber was used [18]. To treat the sample, 100 mL of the UPB solution was pumped onto the top of the sample column. The empty cup was then placed at the bottom of the sample so that any effluent solution that penetrated the sample could be collected and recirculated for 3 h. A new solution of urea and calcium chloride was then introduced through pumping, and this was recirculated for 9 h. These cycles were repeated twice daily with fresh solutions for seven days.

## 3. Testing Methods

### 3.1. Measurement of Calcite

To measure the CC, 5 g of cemented sand was removed from the approximate center of each test column and subjected to a latter elution-analysis measurement. Each sample was dried in an oven at 105 °C and then crushed via hammer tapping to ensure the full acidic solubilization of the calcite. The crushed samples were washed with 2.0 M of HCl solution in order to dissolve the precipitated calcites. This was followed by a final rinse and drain cycle to ensure full calcium extraction. The eluted solution was then placed back in the oven at 105 °C. The difference in the dried weight of the original and eluted-dried samples was taken to be the weight of the calcite precipitated in the sample [21,22].

### 3.2. Testing Procedures

Once the calcite precipitation process was completed, a specimen was washed and saturated with distilled water for one day. Then, the hydraulic conductivity of the specimen was tested using a constant head method. After the hydraulic conductivity test, triaxial measurements were made using a consolidated undrained triaxial compression test under 50, 100, and 300 kPa of confining pressure at a constant axial strain rate of 0.05%/min. Following these tests, the CC of each sample column was measured from pieces of cemented sand removed from the center of each specimen. Finally, scanning electron microscopy (SEM) analysis was carried out to examine the structure of the calcite precipitated sand with and without fiber.

## 4. Results

Two series of undrained triaxial and permeability tests were carried out to investigate the effects of calcite treatment and fiber inclusion. The first series compared three CC treatment ratios of 0%, 2%, and 4%. The second series tested four different fiber contents of 0%, 0.2%, 0.4%, and 0.6% on a specimen with a CC of 4%. In order to ensure the accuracy of the data, only results with an error of less than 0.3% of calcite contents were used. If the CC accuracy was outside this range, another sample was prepared and tested. The test numbers were named with the first two letter-numbers referring the fiber contents (e.g., F0: 0.0% of fiber contents), the middle four letter-numbers referring the confining pressure (e.g., C100: 100 kPa of confining pressure), and the last bracket indicating the calcite content (e.g., (4): 4% of calcite content).

### 4.1. Effect of Calcite Treatment

Table 2 summarizes the testing conditions and compares the peak values and hydraulic conductivity. Figure 2 shows a typical result of the undrained triaxial test on sand with and without calcite treatment and a confining stress of 50 kPa. Deviatoric stress and mean effective stress are defined as $q = \sigma'_1 - \sigma'_3$ and $p' = (\sigma'_1 + 2\sigma'_3)/3$, respectively. The stress–strain curve for untreated sand showed a continuous hardening behavior, but for treated sand showed stiffer behavior at the initial loading stage. A positive excess pore pressure was generated initially, but this became negative due to dilation regardless of the CC. As the CC increased, the dilation decreased due to the cemented particles. O'Donnell and Kavazanjian [13] obtained similar results for biocement with MICP. The relationship between the stress ratio ($q/p'$) and axial strain is shown in Figure 2. The ratio for the untreated sand reached a peak value and then remained constant. On the other hand, the ratio for the treated sands reached a peak and then decreased slightly. The trend for the principal stress ratio ($\sigma'_1/\sigma'_3$) was similar to that of the stress ratio, as shown in Figure 2. This kind of stress–strain behavior has also been observed [23].

**Table 2.** Triaxial test results for samples with different calcite contents.

| Test ID | Average Calcite Content (CC) (%) | Fiber Content (%) | Confining Pressure (kPa) | Hydraulic Conductivity (m/s) | Peak Deviatoric Stress (kPa) | Peak Pore Pressure (kPa) | Peak Stress Ratio | Peak Principal Stress Ratio | CC (%) |
|---|---|---|---|---|---|---|---|---|---|
| CSC50 | | | 50 | $9.03 \times 10^{-3}$ | 725 | $-219.1$ | 1.45 | 3.8 | 0.0 |
| CSC100 | 0.0 | 0.0 | 100 | $9.09 \times 10^{-3}$ | 876 | $-207.6$ | 1.47 | 3.9 | 0.0 |
| CSC300 | | | 300 | $9.01 \times 10^{-3}$ | 1363 | $-193.2$ | 1.46 | 3.8 | 0.0 |
| F0C50(2) | | | 50 | $9.32 \times 10^{-5}$ | 793 | $-150.2$ | 2.54 | 17.6 | 2.1 |
| F0C100(2) | 2.1 | 0.0 | 100 | $9.87 \times 10^{-5}$ | 876 | $-102.2$ | 2.21 | 9.4 | 2.2 |
| F0C300(2) | | | 300 | $9.50 \times 10^{-5}$ | 1362 | $-92.6$ | 1.81 | 5.6 | 2.0 |
| F0C50(4) | | | 50 | $4.07 \times 10^{-5}$ | 874 | $-107.4$ | 2.61 | 20.9 | 4.0 |
| F0C100(4) | 4.0 | 0.0 | 100 | $5.59 \times 10^{-5}$ | 975 | $-68.6$ | 2.38 | 12.5 | 3.8 |
| F0C300(4) | | | 300 | $4.59 \times 10^{-5}$ | 1575 | $-48.8$ | 2.00 | 7.0 | 4.1 |

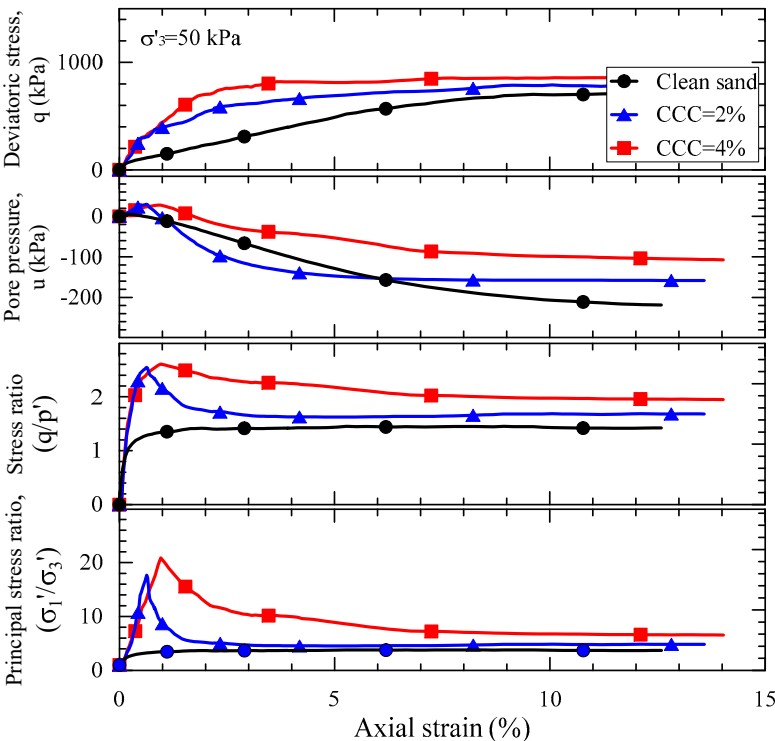

**Figure 2.** Typical results of undrained triaxial tests with different CCs.

Figure 3 shows the relationship between the peak and peak principal stress ratios, and the confining pressure. Untreated sand had almost constant values of peak stress ratio (q/p′ = 1.4) and peak principal stress ratio ($\sigma'_1/\sigma'_3$ = 4), regardless of the confining pressure. These peak values increased as the CC increased. On the other hand, both peak stress ratios of the treated sands decreased as the confining pressure increased. This is because the cementation, caused by the calcite precipitated between the sand particles, is destroyed as the confining pressure increases. If the confining pressure becomes much higher, the treated sand may show similar behavior to the untreated sand.

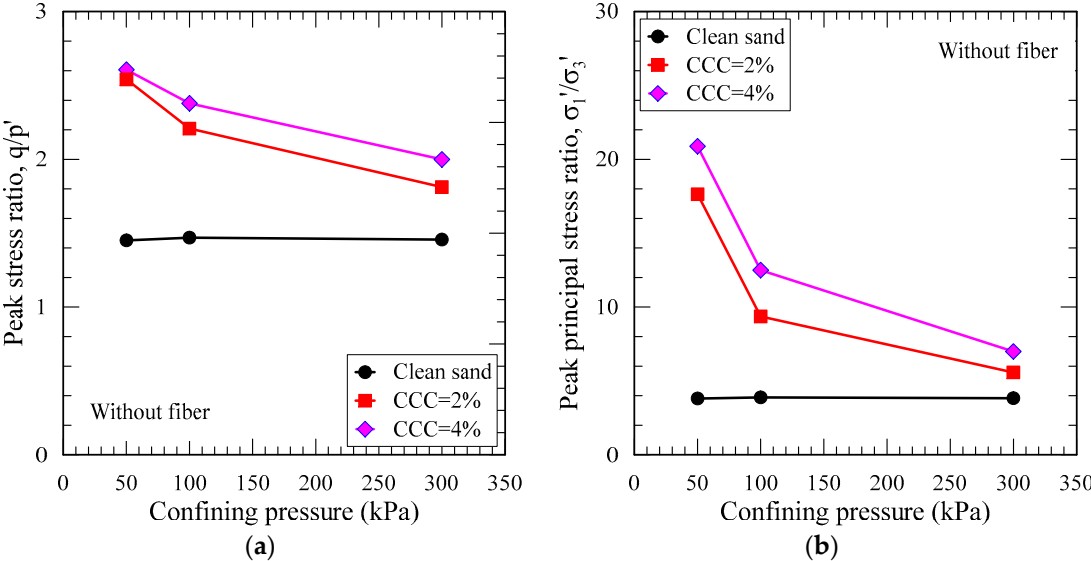

**Figure 3.** Peak stress ratio and peak principal stress ratio for samples with different CCs against confining pressures: (**a**) peak stress ratio; (**b**) peak principal stress ratio.

Figure 4a shows the Mohr's stress circles at different confining pressures. The friction angle and cohesion were obtained from the Mohr's stress circles and compared for different CCs. Table 3 summarizes these strength parameters. As the CC increased, both the cohesion and friction of the treated sand increased due to the cementation of sand particles [23,24]. In general, the friction angle of sand can be influenced by the relative density, particle size, and shape of the sand. In addition to the cementation effect, the shape of the sand became angular due to precipitated calcite, which may also have contributed to the increased shear strength.

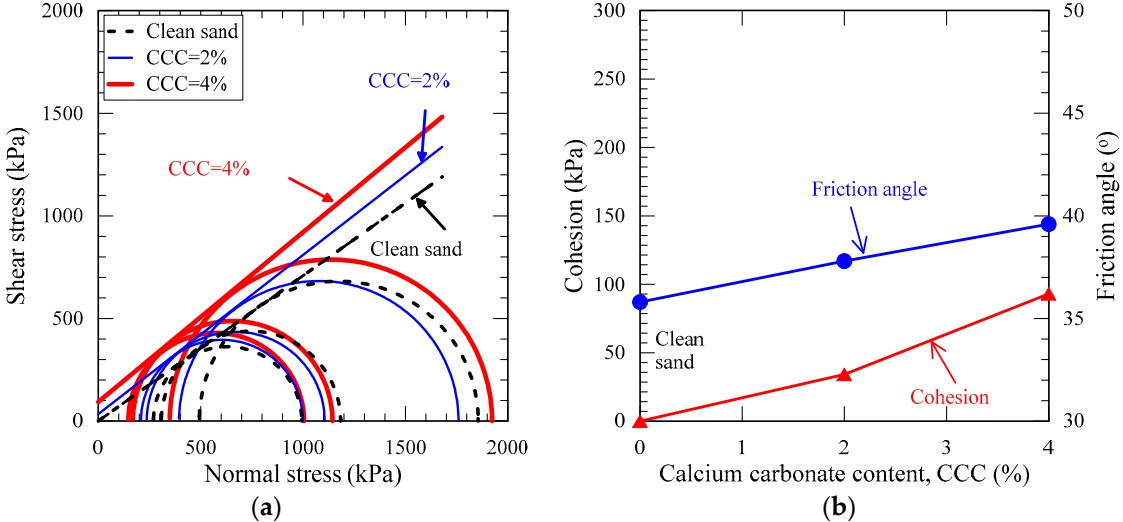

**Figure 4.** Engineering properties of Ottawa sand treated by MICP: (**a**) Mohr's stress circles; (**b**) cohesion and friction angle.

**Table 3.** Strength properties of samples with different CCs and without fiber.

| Type | Calcite Content (CC) (%) | | |
|---|---|---|---|
| | 0 | 2 | 4 |
| Cohesion (kPa) | 0 | 34.2 | 93 |
| Friction angle (°) | 35.3 | 37.8 | 39.6 |

The hydraulic conductivity of clean Ottawa sand is approximately $9.0 \times 10^{-3}$ m/s. These values dramatically decreased by a factor of 100 as the CC increased up to 4%. Other researchers [4,5] have produced similar results. This is attributed to the reduction in the voids between the sand particles, which is due to the precipitated calcite.

*4.2. Effect of Fiber Inclusion*

The effect of fiber inclusion on shear strength and permeability was described in this section. The results for sand treated with 4% CC and with four different fiber contents (FC = the weight of fiber/the weight of dry sand) (0.0%, 0.2%, 0.4%, and 0.6%) are summarized in Table 4.

**Table 4.** Triaxial test results with 4% CC and different fiber contents.

| Test ID | Average CC (%) | Fiber Content (%) | Confining Pressure (kPa) | Hydraulic Conductivity (m/s) | Peak Deviatoric Stress (kPa) | Peak Pore Pressure (kPa) | Peak Stress Ratio | Peak Principal Stress Ratio | CC (%) |
|---|---|---|---|---|---|---|---|---|---|
| F0C50(4) | | | 50 | $4.07 \times 10^{-5}$ | 874 | $-107.4$ | 2.61 | 20.9 | 4.0 |
| F0C100(4) | 4.0 | 0.0 | 100 | $5.59 \times 10^{-5}$ | 975 | $-68.6$ | 2.38 | 12.5 | 3.8 |
| F0C300(4) | | | 300 | $4.59 \times 10^{-5}$ | 1575 | $-48.8$ | 2.00 | 7.0 | 4.1 |
| F2C50(4) | | | 50 | $3.67 \times 10^{-5}$ | 1085 | $-95.2$ | 2.62 | 21.5 | 4.4 |
| F2C100(4) | 4.0 | 0.2 | 100 | $3.83 \times 10^{-5}$ | 1271 | $-72.3$ | 2.44 | 14.0 | 3.9 |
| F2C300(4) | | | 300 | $5.60 \times 10^{-5}$ | 1942 | $-57.6$ | 1.95 | 6.6 | 3.8 |
| F4C50(4) | | | 50 | $3.13 \times 10^{-5}$ | 1199 | $-107.7$ | 2.60 | 20.7 | 4.2 |
| F4C100(4) | 4.1 | 0.4 | 100 | $5.55 \times 10^{-5}$ | 1270 | $-74.5$ | 2.50 | 16.0 | 3.9 |
| F4C300(4) | | | 300 | $3.18 \times 10^{-5}$ | 2007 | $-64.5$ | 1.96 | 6.6 | 4.3 |
| F6C50(4) | | | 50 | $2.46 \times 10^{-5}$ | 1213 | $-107.1$ | 2.61 | 21.2 | 4.2 |
| F6C100(4) | 4.2 | 0.6 | 100 | $2.92 \times 10^{-5}$ | 1445 | $-95.9$ | 2.49 | 15.7 | 4.3 |
| F6C300(4) | | | 300 | $3.94 \times 10^{-5}$ | 2253 | $-77.5$ | 2.04 | 7.4 | 4.1 |

Figure 5 shows a typical result for the undrained triaxial test with different fiber contents (FCs) and a confining pressure of 300 kPa. The deviatoric stress increased continuously as the axial strain increased, regardless of the fiber contents. The peak deviatoric stress increased as more fiber was included. Consoli et al. [24] found similar results for reinforced cemented sand with 0.5% fiber. An excess pore pressure showed less dilative behavior as the fiber content increased. The relationship between the stress ratio and the axial strain showed a similar pattern to treated sand without fiber. The peak principal stress ratio increased as the cementation ratio and fiber content increased, and it decreased as the confining pressure increased [25,26].

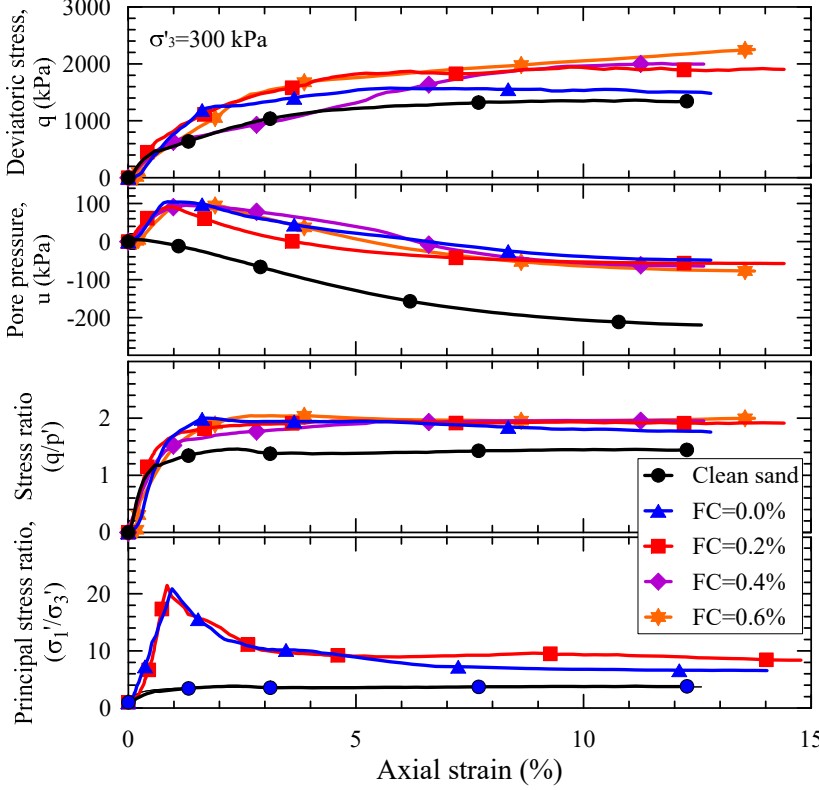

**Figure 5.** Typical results of undrained triaxial tests with different fiber contents.

Figure 6 shows the relationship between the peak stress ratio and peak principal stress ratios versus the confining pressure. The peak stress ratios of the treated sand with fiber were much higher than that of the sand without fiber. The peak stress ratios were less affected by the fiber content.

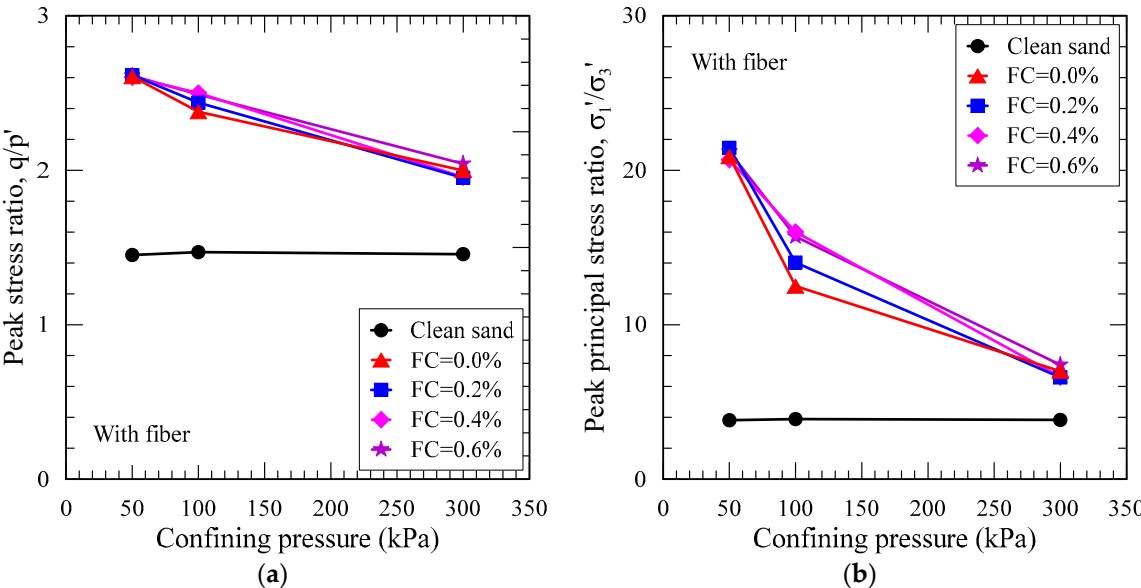

**Figure 6.** Peak stress ratio and peak principal stress ratio for samples with different fiber contents against confining pressure: (**a**) peak stress ratio; (**b**) peak principal stress ratio.

Figure 7a shows the Mohr's stress circles and failure envelope for the treated sand with and without fiber under three confining pressures. From these stress circles, the cohesion and friction angle were calculated and are compared in Figure 7b and Table 5. Both the cohesion and friction angle increased as the fiber content increased. The strength gradually increased up to a fiber content of 0.2% and then it remained constant. In this study, the optimum fiber content was approximately 0.4%. Choi et al. [18] discovered that the UCS rapidly increased as the PVA fiber content increased up to 0.8% when the CC was high. Li et al. [17] showed that the shear strength of MICP treated sand with polypropylene fiber increased up to a ratio of 0.3%. Therefore, the optimum fiber content for sand treated by MICP could be influenced by the CC and testing methods.

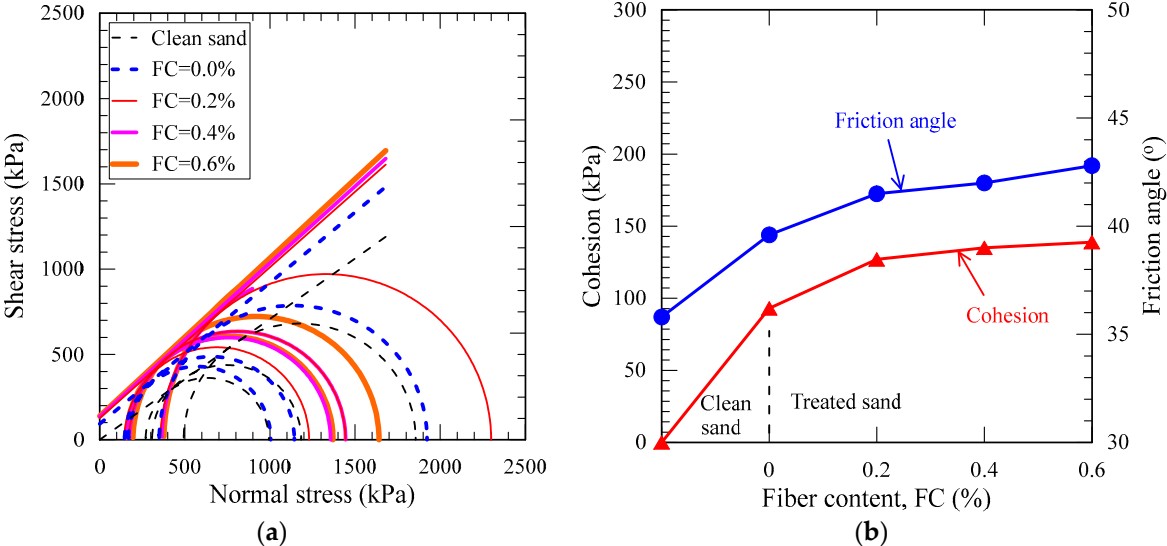

**Figure 7.** Engineering properties of Ottawa sand treated by MICP and fiber: (**a**) Mohr's stress circles; (**b**) cohesion and friction angle.

**Table 5.** Strength properties of 4% CC samples with different fiber contents.

| Type | Fiber Content (%) | | | | |
|---|---|---|---|---|---|
| | Clean Sand | 0 | 0.2 | 0.4 | 0.6 |
| Cohesion (kPa) | 0 | 93 | 127 | 135 | 139 |
| Friction angle (°) | 35.3 | 39.6 | 41.5 | 42 | 42.8 |

As the fiber content increased, the average hydraulic conductivity decreased slightly due to the reduction in the void volume caused by the fiber and calcite within the sand particles. However, the effect of fiber inclusion on permeability appears to be minor compared to the precipitation of calcite [18].

*4.3. Microstructure of Biocementation with Fiber*

Following the triaxial tests, part of one sample was retrieved and analyzed using SEM. Figure 8 shows the SEM image for 150 and 1500 times magnification. The images show the distribution and cubic shape of the precipitated calcite surrounding the sand particles and fiber. This bonding may contribute to the improvement in the engineering properties [17,18]. The calcite particles were less than 10 μm in diameter, as shown in Figure 8b, which may affect the strength of the biocemented sand [5].

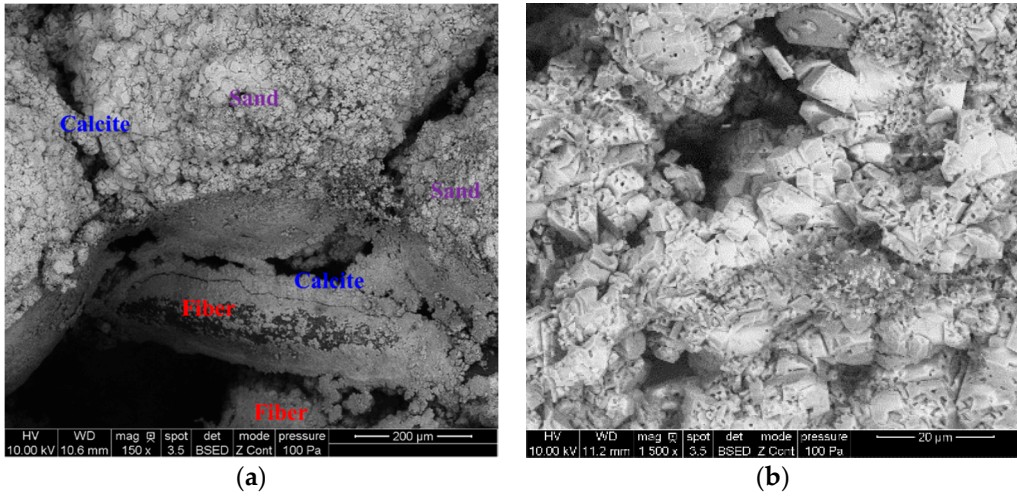

**(a)**        **(b)**

**Figure 8.** SEM images of the sample with 0.6% fiber under different magnifications: (**a**) 150 times magnification; (**b**) 1500 times magnification.

## 5. Conclusions

Clean Ottawa sand was treated with CCs of 0%, 2%, and 4% and fiber contents of 0%, 0.2%, 0.4%, and 0.6%. Its undrained behavior was investigated in terms of cohesion and friction angle, and its hydraulic conductivity was also examined. The test results are summarized as follows:

(1) As the CC increased from 0% to 4%, the friction angle and cohesion increased from 35.3° to 39.6° and from 0 to 93 kPa, respectively. This is mainly attributed to the precipitated calcite, which caused the cementation of the sand particles and made the particles more angular.

(2) For a specimen with a CC of 4%, as the fiber content increased from 0% to 0.6%, the friction angle and cohesion increased from 39.6° to 42.8° and from 93 to 139 kPa, respectively. This can be explained by the fiber bridging of each sand particle.

(3) The hydraulic conductivity of clean Ottawa sand decreased by a factor of more than 100 as the CC increased from 0% to 4%. This is because the calcite that precipitated within the sand particles

reduced the void volume. On the other hand, the amount of fiber inclusion only had a minor effect on the hydraulic conductivity of 4% calcite precipitated sand because it already had a dense structure.

**Author Contributions:** Conceptualization, S.-G.C. and T.H.; Methodology, S.-G.C. and S.-S.P.; Formal Analysis, S.-G.C. and T.H.; Writing—Original Draft Preparation, S.-G.C.; Writing—Review and Editing, S.-S.P.; Supervision, S.-S.P.

**Funding:** This work was supported by the National Research Foundation of Korea (NRF) grant funded by the Korea government (No. NRF-2018R1A5A1025137).

**Conflicts of Interest:** The authors declare no conflict of interest.

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
