# Peer review of "Undrained Behavior of Microbially Induced Calcite Precipitated Sand with Polyvinyl Alcohol Fiber"

_applsci, doi:10.3390/app9061214_

Reviewer 1 Report

This paper presents the triaxial test results on MICP treated sand with varying fiber contents. The research is well designed, and the way of analyzing and presenting the testing results is acceptable for journal publication. There are a few arguments the reviewer raised, but the authors will find appropriate answers to them. The reviewer recommends this article to the publication after minor revisions. The reviewer hope the authors find the comments helpful to improve the paper.

Title:

The authors may name the MICP as Microbially Induced Calcium Carbonate Precipitation, and I understand there may exist a philosophy in the terminology.  Nevertheless, in my opinion, the author is recommended slightly modifying the calcium carbonate into calcite as previous researches. Otherwise, the MICP should be changed to MICCP throughput the paper.

Abstract

CCCs of 0, 0.2, 0.4% -> Is this 0, 2, 4%?

Introduction

Line 33; Microbially Induced Calcite Precipitation (MICP)

Line 42: CaCO3=> typo. subscript

Sample preparation

Line 70: Use reference for the ASTMs 

Line 75: A PVA fiber -> PVA fibers shown in Figure 1(b) were used . 

Line 84: PVA fiber -> PVA fibers

Line 96: dry unit weight of 16.19 -> what is the relative density of the clean Ottawa sand at the initial condition? Relative density more explains the state of sand to readers on testing.

Line 114: How did you wash the specimen? Is it by circulating the distilled water through the specimen? what did you expect to wash away out of the specimen? Do you expect any effect of removal of residues in void? Clarify that the triaxial testing was performed on the identical specimen after the permeability test.

Line 127: The "erratic" results were removed in the analysis. How do you define the "error"? Is the error from the trend line? Is it in stress (kPa)? Is the 0.3% reasonable criteria to exclude some results? Isn't it too strict?

Table 2: Test ID may include testing conditions. Explaining the meaning of each symbol would be very helpful to readers.

Use m/s instead of cm/s. Scientists are in the move of eliminating cm in scientific writing as every units are defined in the order of 10^3 

Line 146: It is unclear how did you define "peak" value from triaxial tests. It is very important as the axial strain extends nearly up to 15%. 

Line 153: So the contribution of void filling of calcium carbonates diminishes at higher confining pressure. In that case, why the friction angle increases with higher CCC? Even with high CCC, the peak deviatoric stress will be eventually similar at high confining pressure. Then, the friction angle would decrease (while the cohesion has significantly improved) at higher CCC. Needs explanation.

Figure 8: fiber inclusion was not observed in the SEM. Please use captions to indicate which is which.

Reviewer 2 Report

In general the paper was well written and easy to understand. However on every point where results are presented the author notes the similarity of the results to those of other authors e.g.

line 137 "O'Donnell and Kavazanjian (2015) obtained similar results".

line 142 "This kind of stress-strain behaviour was also observed by Montoya and Dejong (2015)".

line 160 the key observation, and one of the conclusions of the paper is referenced to another author "As the CCC increased, both the cohesion and friction of the treated sand increased due to the cementation of sand particles [22-23]".

line 171 "Other researchers have produced similar results"

line 182 "Consoli et al found similar results

line 185-186 the key finding is referenced to another author similar to line 160

line 201 the optimum fibre content of the study is identified and again similar values are brought in from other studies

line 212, 217 and 219 again key findings of the paper are referenced to other authors

The authors need to identify the novel findings of the paper and make them more prominent, where references are made to studies which will be slightly different, it may be useful for the authors to highlight this difference to show that their results are novel in addition to being in the correct range. Where the authors have results and make statements which are based on their results, the statements should ne be referenced to others.

The referencing of the article needs to be looked at, I don't believe any of the in text citations need the year? Some of the references have the year twice, number 3, 5, 11, 23, 24, 25. There is an "&" in the list of authors in some citations and not in others e.g. number 10. Some journals are abbreviated, some are not. It would be worth the author checking through the references and merging them to a common style.

A few other small points

Line 18, the percentages should be 0%, 2% and 4%

Line 20 delete the "and" after the bracket

Line 75 should be "PVA fibre" unless only one fibre was used, also "it was a" should be "it has a"

For figure 1a, why is the grain size x-axis reversed. perhaps this is just something of preference and it does not affect the results, but I would draw the axis the other way

Line 92 "rand" should read "range"
